

# Characteristics and development of steepland gullies in the dry valleys of Southwest China

Tingting Cui[1,2,3], Yuli He[1,2,3], Lei Wang[1,2,3], Jun Luo[1,2,3], Ting Xiao[1], Hui Liu[1,2,3], Bin Zhang[1,2,3], Qingchun Deng[1,2,3] and Haiqing Yang[1,2,3]

[1] Sichuan Provincial Engineering Laboratory of Monitoring and Control for Soil Erosion in Dry Valleys, China West Normal University, Nanchong, China
[2] Liangshan Soil Erosion and Ecological Restoration in Dry Valleys Observation and Research Station, Xide, China
[3] School of Geographical Sciences, China West Normal University, Nanchong, China

Corresponding author
Yuli He, heyuli301@126.com

## ABSTRACT

In semi-arid and arid areas, gully erosion is one of the most destructive forms of erosion and causes serious land degradation and resource destruction. Steepland gullies are widely distributed in the dry valleys of southwest China, and their formation is one of the main causes of soil erosion and the destruction of sloping farmland in the region. Previous research on the development of steepland gullies is limited, and further study is needed. In this study, 11 steepland gullies at various stages of development located in Guobu Village, Xide County, Liangshan, Sichuan Province, were selected for investigation using a digital elevation model (DEM) derived from unmanned aerial vehicle data as the primary data source. These data had a spatial resolution of 0.1 m. Fundamental parameters such as the gully length, width, depth, area, and volume were extracted from the remote sensing data. Other characteristic parameters, including the coefficient of main and tributary gullies, vertical gradient, gully elongation, and gully openness, were also investigated. The results indicate a significant linear positive correlation between the gully's degree of openness and elongation as the gully's length, width, and depth increase. Furthermore, the vertical gradient and coefficient of main and tributary gullies exhibit power-law relationships with these gully dimensions. The development of steepland gullies was divided into infancy, youth, maturity, and old age based on the use of the gully length as an ergodic indicator in space-for-time substitution. The morphological characteristics of these different stages were quantitatively analyzed, and a proposed mechanism for how the evolution of the gullies proceeds was developed. An empirical model of volume–length erosion was established to investigate the development process of steepland gullies in the dry valleys. It has been observed that the development law of steepland gullies is essentially consistent with the very active stage of typical gully formation, suggesting that steepland gully may represent the initial stage of gully development. The results show that these steepland gullies have their origin in high-intensity rainfall events that are accompanied by the formation of steps and drop water. The effects of gravity erosion and hydraulic erosion then cause the gullies to expand rapidly, forming gullies with a large head and a small tail before they gradually stabilize. The results of this study will help with the understanding of the formation and evolution of steepland gullies and will be of practical significance for

the prevention of gully erosion and the protection of sloping farmland in the dry valley region of southwest China.

# INTRODUCTION

Soil erosion is one of the major global environmental problems that endanger human survival and development (*Castillo & Gómez, 2016*; *Ionita et al., 2021*; *Wuepper, Borrelli & Finger, 2020*). Gully erosion, which destroys sloping farmland and leads to significant soil and water loss, is one of the most important forms of soil erosion and one of the most serious manifestations of soil degradation (*Bewket & Sterk, 2003*; *Bruno, Stefano & Ferro, 2008*; *Le Roux & Sumner, 2012*; *Nearing et al., 1997*). In certain loess regions of Europe, gully erosion contributes an average of over 30% to the overall soil loss (*Casalí, Giménez & Bennett, 2009*; *Poesen et al., 2003*), while in China, it accounts for half of the total watershed loss (*Wu et al., 2018*). The rate at which permanent gullies cause soil loss markedly surpasses that of cultivated land and stands as the primary driver behind global landscape degradation (*Allen et al., 2018*; *Bennett & Wells, 2019*). The dry valley region of southwest China has a fragile ecological environment that suffers from severe soil erosion, broken surfaces, and active gully erosion (*Zheng & Gao, 2003*). In the dry valley region of southwest China, slopes with a gradient exceeding 25° comprise 55% of the total area (*Fan et al., 2020*). The steepland gully is a distinctive erosion gully type found in the dry valleys of the southwest region. It represents a permanent feature formed on slopes with a notable incline, characterized by its wide body and tapering tail. As the erosion gully progresses, the tail gradually thins out and either merges into adjacent gullies or dissipates entirely on the slope, resulting in an overall spoon-shaped or palm-shaped morphology with distinct gully patterns. The continuous expansion of steepland gullies has led to the ongoing depletion of farmland resources, resulting in reduced land productivity and severe negative impacts on local livelihoods, agricultural production, and regional economic development. An accurate depiction of the processes involved in gully erosion is essential for comprehending the underlying mechanisms, and an investigation into the development of steepland gullies can help with the effective protection of sloping farmland while also playing a vital role in the preservation of water and soil resources.

Gully evolution has always been a prominent topic in the field of gully geomorphology (*Bewket & Sterk, 2003*; *He et al., 2018*; *Jiang et al., 2018*; *Ran et al., 2018*; *Roberts & Gregg, 2019*) and is influenced by various factors, including the basin area, surface runoff, precipitation index, previous precipitation, soil moisture, and piping index (*Berger et al., 2010*; *Seginer, 1966*). Due to the diverse internal and external forces that influence the different stages of gully development, notable spatiotemporal variations are observed (*Gales et al., 2013*), making an accurate description of gully erosion essential. Gullies constitute a significant geomorphic unit of dry valleys and serve as a crucial conduit for sediment

and pollutant transport (*Zhang et al., 2022*). The presence and developmental processes of gullies markedly influence the geomorphic characteristics and subsequent evolution of the landscapes where they occur (*Zhang, 2020*). Although the formation of gullies is relatively rapid, their subsequent development progresses comparatively slowly.

Previous monitoring of gully erosion revealed that a series of interconnected processes, including leakage, soil creep, collapse, and erosion, regulate gully development (*Sidorchuk, 1999*; *Soufi, 2002*; *Zheng, Xu & Qin, 2016*). The primary mechanisms contributing to gully development are headwater erosion, lateral erosion of the gully slope, and cutting erosion beneath the gully (*Jing, 1986*). Its formation and evolution are influenced by various factors (*Shen et al., 2015*), including topography, parent material and soil properties, climate conditions, hydrological processes, vegetation coverage, and current land-use changes (*Hayas, Poesen & Vanwalleghem, 2017*; *Parkner et al., 2006*; *Rahmati et al., 2022*; *Torri et al., 2018*). The formation and development of spoon gullies in loess areas result from the interaction between external factors such as climate, soil and vegetation and internal factors such as geology and geomorphology (*Li et al., 2022*). In the subtropical and semi-humid regions of Tennessee, United States, gully incision and expansion are closely correlated with antecedent rainfall, cumulative rainfall, and rainfall duration (*Luffman, Nandi & Spiegel, 2015*). Agricultural activities and disturbance intensity, land reclamation for farming purposes (*Chaplot et al., 2011*; *Lesschen et al., 2007*; *Saxton et al., 2012*; *Selkimäki & González-Olabarria, 2017*), overgrazing, deforestation through forest cutting practices, road construction activities, and urbanization markedly influence gully incision initiation and progression. Forest cover markedly contributes to inhibiting gully incision development (*Fajardo, Llancabure & Moreno, 2022*; *Nogueras et al., 2000*; *Vandekerckhove et al., 2000*). Basic geometric characteristics such as gully network density, tortuosity, cutting degree, and inclination angle are effective when evaluating morphological evolution in river landforms (*Bryan & Rockwell, 1998*; *Shen et al., 2015*), whereas parameters such as length, width, and depth are efficient for characterizing steepland gully evolution (*Deng et al., 2015*; *Wang et al., 2023*). The coefficients of main and tributary gullies, gully openness degree, vertical gradient, gully elongation, and the length-to-width ratio can directly quantify erosion gully development (*Liu et al., 2016*). Examining the correlation between these fundamental indicators can reveal steepland gully development patterns and provide insights into the mechanisms underlying erosion gully formation (*Berger et al., 2010*; *Kimaro et al., 2008*).

Research on steepland gullies has been conducted worldwide, including in New Zealand and Ethiopia, with a primary focus on the factors that influence the development of the gullies (*Betts, Trustrum & Rose, 2003*; *Billi & Dramis, 2003*; *Frankl et al., 2013a*; *Marden et al., 2012*). Steepland gullies are widely distributed in the dry valley regions. However, research on its development and evolution process remains in the preliminary exploration stage. While steepland gully development disrupts land resource integrity and effectiveness, it can also exacerbate soil erosion on slopes. Therefore, the study of steepland gullies in the dry valleys is of considerable importance. In this study, using space-for-time substitution, steepland gully development was divided into stages based on spatial factors. The characteristics and erosion mechanisms at each stage were then used to develop a proposed model for the progression of gully evolution. This approach will

aid in understanding steepland gully formation and evolution while providing a reliable theoretical foundation for predicting gully erosion and preventing soil erosion.

## STUDY AREA

The Liangshan region is acknowledged as one of the regions of China that suffers most seriously from soil erosion (*Liu & Zhao, 2004*). The region has a subtropical monsoon climate characterized by an average annual temperature of 14 degrees centigrade, average annual precipitation of 1,006 mm, and average annual evaporation of 1,945 mm. The precipitation is not evenly distributed throughout the year. Purple and yellow-brown soil types predominate, and low-to-medium-altitude mountains constitute approximately 70% of the land area. This study focused on Guobu Village, which is located in Xide County of Liangshan of Sichuan Province and features topography where the elevation gradually decreases from north to south and east to west; the elevation has a range of approximately 1,680–2,025 m. This area is located on the east side of the Anning River–Zemu River fault zone (Fig. 1A), which is the principal active fault at the eastern boundary of the Sichuan–Yunnan active block. The robust tectonic background is influenced by the collision and compression between the Indian and Eurasian plates, which creates a complex geological environment characterized by well-developed secondary fault structures, extensive rock fragmentation, and poor stability (*Jiang et al., 2015*; *Replumaz et al., 2001*; *Wang et al., 1998*). Notably, numerous erosion gullies, including steepland gullies at various stages of evolution, are widely distributed across the area and comprise a relatively comprehensive evolution sequence that facilitates the investigation of steepland gully development and evolution (Fig. 1B).

## DATA AND METHODS

### Data collection

A Pegasus D200S multi-rotor unmanned aerial vehicle (UAV) equipped with a LiDAR110 lidar was used to carry out a digital terrain survey of the study area on January 14th, 2022. Eleven field control points were set in conjunction with real-time kinematic positioning for precise calibration of both the vertical and horizontal data. The UAV Butler software was utilized to convert the lidar data format, and the laser point cloud trajectory was calculated using the Inertial Explore high-precision GNSS/INS post-processing software. Subsequently, the point cloud data underwent calculation using the UAV Butler software. The data were then segmented into blocks, followed by noise removal and classification; this ultimately produced a digital elevation model (DEM) with a resolution of 0.1 m, along with a processing flow chart for the laser point cloud data (Fig. 2).

### Methods

#### Use of spatiotemporal ergodic indicators for studying the development of steepland gullies

When utilizing the space-for-time substitution method to investigate the evolution of river geomorphology, it is necessary to ensure that the selected experimental objects are situated at an equivalent watershed level and have comparable watershed morphologies; in other

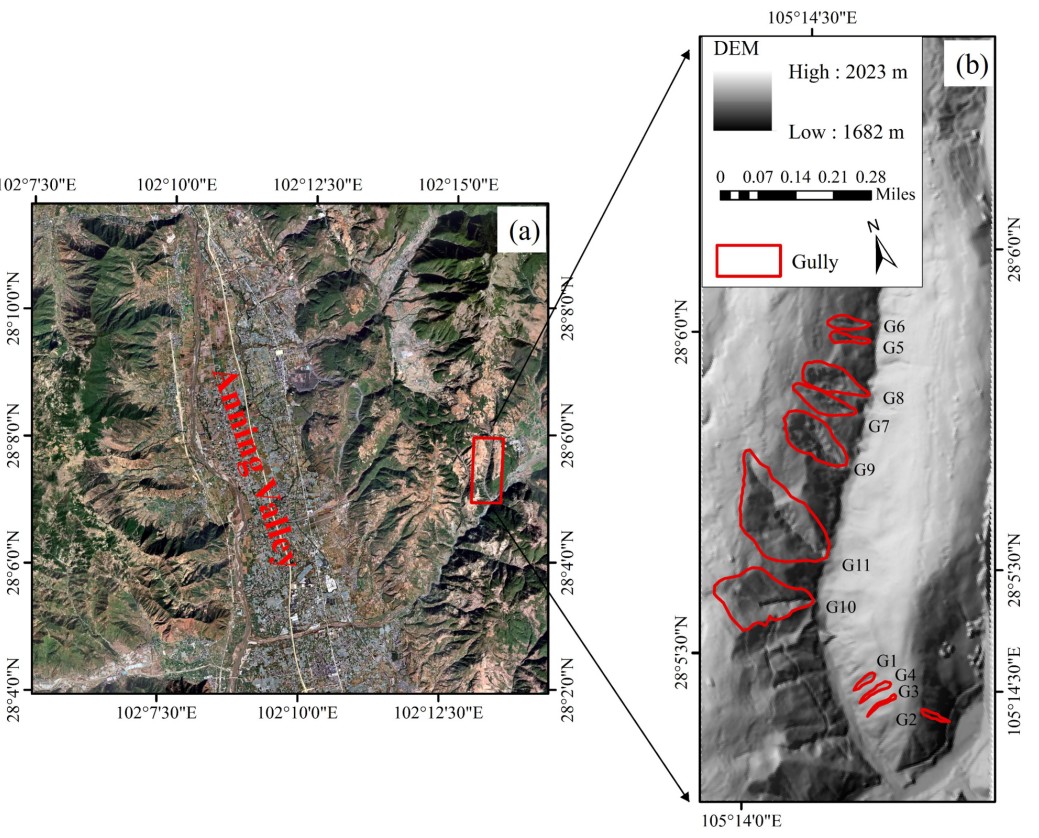

**Figure 1** **The study area.** (A) The data set is provided by Geospatial Data Cloud site, Computer Network Information Center, Chinese Academy of Sciences; Landsat 8 OLI_TIRS Satellite Digital Products (https://www.gscloud.cn/sources/accessdata/411?pid=263).

words, the chosen experimental subjects should share a similar or identical developmental environment (*Fryirs, Brierley & Erskine, 2012*). In this study, we selected a representative region featuring steepland gullies within a dry valley. This area exhibits well-developed geological structures, steep terrain gradients, concentrated precipitation patterns, and sparse vegetation cover, as well as fragmented rock and soil compositions; these provide the basic conditions necessary for the formation of erosion gullies. Moreover, various stages of steepland gully evolution have been preserved within the study area, thereby forming a relatively comprehensive sequence.

Space-for-time substitution often employs ergodic indicators to reflect the spatiotemporal development of individual landforms and thus represent the changing characteristics of research objects. In studies on the evolution of geomorphology, many researchers have utilized morphological measurement parameters such as distance, location, and geomorphologic dimension or complexity as ergodicity indices (*Huang et al., 2019*). *Micallef et al. (2014)*, *Yang et al. (2021b)*, and others used space-for-time substitution to investigate the evolution of gullies and selected the gully length as an ergodic indicator to represent the different stages of gully development. In this study, this

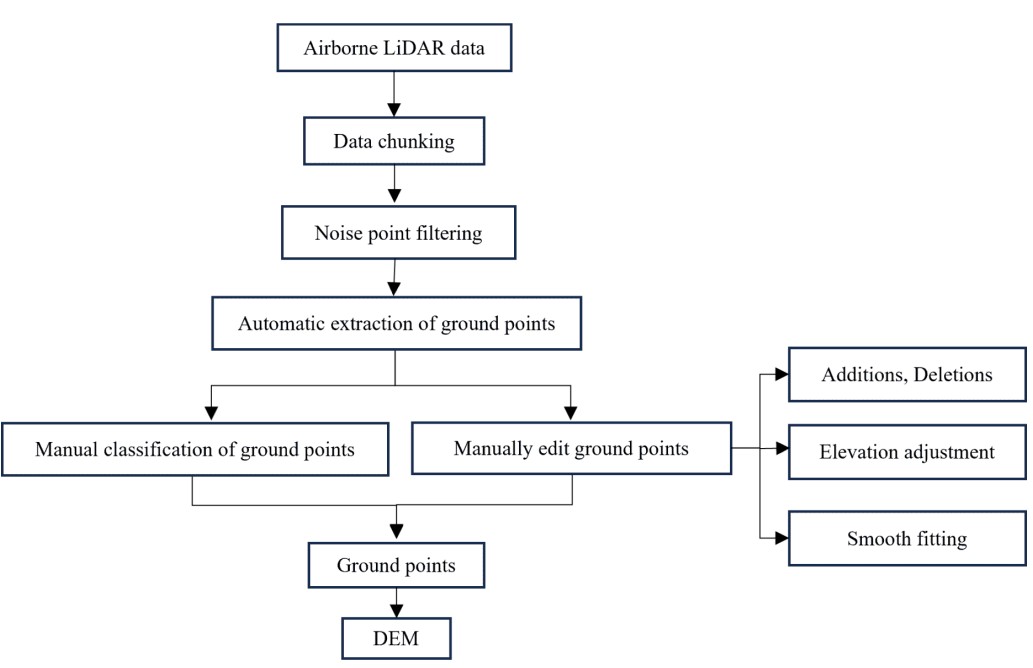

**Figure 2  Flowchart of data processing.**

method was adopted for sorting and numbering the steepland gullies based on the length of the main gully.

### Morphological indicators used to represent the spatiotemporal development of steepland gullies

The selected gully parameters were first extracted using the ArcGIS software to obtain DEM data for the steepland gullies. Referring to earlier studies (*Micallef et al., 2014*), the indicators that can effectively reflect the morphological changes during the development of steepland gullies were selected: the coefficient of the main and tributary gullies, vertical gradient, gully elongation, and gully openness. Descriptions of these indicators are given below. (1) The coefficient of the main and tributary gullies, $R$, is the ratio of the length of the main gully to the total length of the gully:

$$R = \frac{L}{L_0} \tag{1}$$

where $L$ is the length of the main gully, and $L_0$ is the total length of the gully. (2) The vertical gradient, $I$, is the ratio of the difference in elevation between the gully's head and tail to its length:

$$I = \frac{H_2 - H_1}{L} \tag{2}$$

where $H_2$ is the elevation of the gully head, $H_1$ is the elevation of the gully tail, and $L$ is the length of the gully. (3) The gully elongation, $F$, is the ratio of the gully's width to its length:

$$F = \frac{W}{L} \tag{3}$$

where $W$ represents the width of the gully, and $L$ represents its length.

The gully elongation quantifies the relationship between the length and width of the rectangle formed by enclosing the gully along the direction of the normal to the slope. It also indirectly classifies the gully according to the type of tail: according to Huang Jinlv's proportion standard, $F < 0.618$ corresponds to a "slender" tail, $F = 0.618$ corresponds to a tail of the "Huang Jinlv type", and $F > 0.618$ corresponds to a "short, coarse" tail. (4) The gully openness, $K$, is a measure of the topographic openness of the gully:

$$K = \frac{W}{H} = \frac{1000A}{L \cdot H} \tag{4}$$

where $W$ is the width, $A$ is the area, $L$ is the length, and $H$ is the relative height difference of the gully.

A gully with $K > 0.65$ is defined as "open", $0.35 < K < 0.65$ corresponds to a "semi-open gully", and an "incised" gully has $K < 0.35$.

### Steepland gully cross-section indicators

During the process of gully development, the cross-section morphology serves as a significant indicator of gully dynamics. The VF indicator, a measure of the width-to-height ratio of a valley, can be employed to quantitatively assess a gully's cross-sectional characteristics and thus provide insight into regional uplift patterns. Given the specific conditions prevailing in the study area, for each gully, we selected five cross-sections for analysis: these were named upper gully, upper-middle gully, middle gully, lower-middle gully, and lower gully (Fig. 3). The average VF value was then calculated for each cross-section (*Bull & Mcfadden, 1977*). A low VF value (VF < 1.0) was considered to indicate a V-shaped valley with rapid regional uplift and intense downcutting erosion by a river. A high VF value (VF > 1.0) was considered to correspond to a wider valley where tectonic activity is relatively weak and lateral erosion dominates over downcutting erosion. The value of the VF indicator was calculated as

$$VF = \frac{2VFfw}{[(Eld - Esc) + (Erd - Esc)]} \tag{5}$$

where Ffw represents the width of the valley floor, Eld and Erd represent the elevations of the shoulders on the two sides of the valley, and Esc represents the average elevation of the valley floor.

### Empirical volume–length relationship

There is a power function relationship between the volume and length of the erosion trench, which usually closely fits the data set, so the volume–length (V–L) model can be used to predict the volume of the gully (*Frankl et al., 2013b*).

$$V = a \cdot L^b. \tag{6}$$

Here, $V$ represents the volume of the gully, and $L$ represents its length. The higher the absolute value of $a$ is, the stronger the correlation between $V$ and $L$; $b$ reflects the rate at which the volume changes in response to variations in gully length.

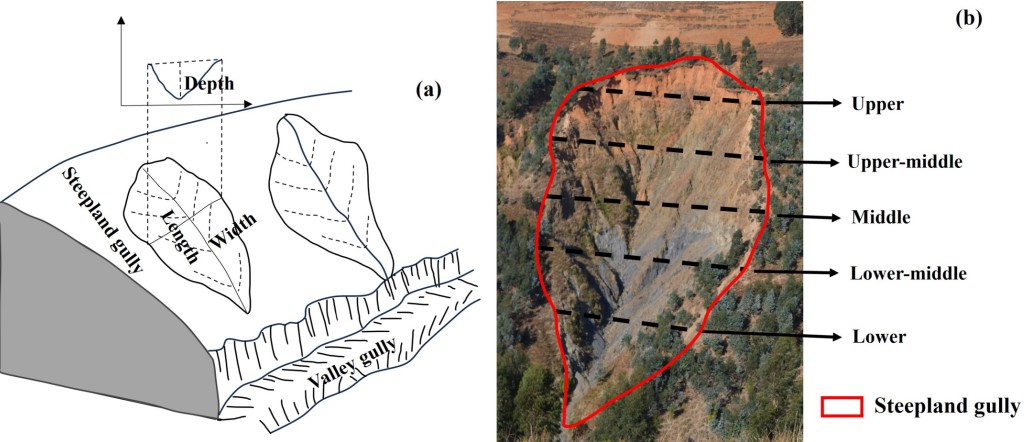

**Figure 3 Schematic of gully profile line locations.** Photo credit: Ting Xiao.

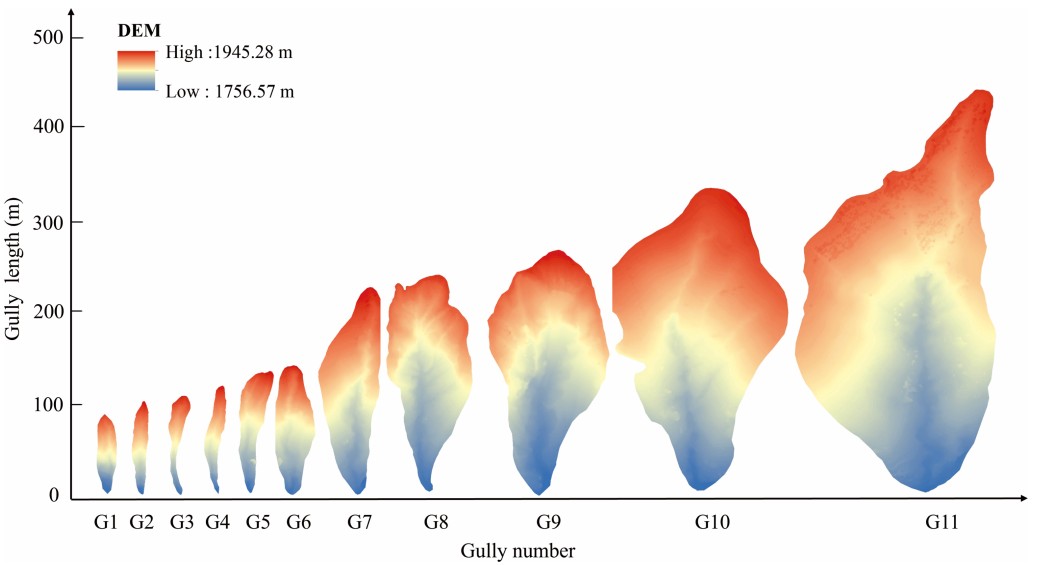

**Figure 4 Sequencing results of steepland gullies.**

## Data processing

In this study, high-resolution DEM data and the space-for-time substitution approach were used to select 11 steepland gullies that were at various stages of development (Figs. 1B and 4). Basic indicators such as the gully length, width, depth, area, volume, vertical gradient, width-to-length ratio, openness, and the coefficient of main and tributary gullies were extracted using ArcGIS software.

**Table 1  Statistical results of basic indicators of steepland gullies in Guobu village.**

| Gully No. | Gully length (m) | Gully width (m) | Gully depth (m) | Area (m²) | Volume (m³) |
|---|---|---|---|---|---|
| G1 | 82.29 | 11.56 | 3.57 | 1159.69 | 4140.09 |
| G2 | 100.44 | 12.96 | 3.88 | 1107.41 | 4291.23 |
| G3 | 114.01 | 11.12 | 3.19 | 667.34 | 2128.81 |
| G4 | 117.69 | 12.01 | 3.68 | 1176.95 | 4331.17 |
| G5 | 141.02 | 19.39 | 4.86 | 2190.96 | 10641.78 |
| G6 | 144.06 | 30.24 | 6.50 | 3604.47 | 23429.08 |
| G7 | 227.94 | 45.61 | 16.29 | 8248.49 | 134332.59 |
| G8 | 239.70 | 61.90 | 24.29 | 11459.63 | 278305.31 |
| G9 | 265.60 | 91.50 | 28.86 | 20630.29 | 595331.27 |
| G10 | 363.87 | 114.11 | 30.57 | 31783.21 | 971658.19 |
| G11 | 498.82 | 145.87 | 41.57 | 50878.91 | 2115087.19 |

# RESULTS AND ANALYSIS

## Morphological characteristics of steepland gully
### Planar characteristics

Based on the lengths of the main gullies, the steepland gullies in the study area were sorted and numbered (Fig. 4); gully G1 was the shortest at 82.29 m, and gully G11 was the longest at 498.82 m. Gully G3 was the narrowest (width = 11.12 m), and gully G11 was the widest (width = 145.87 m). The shallowest gully, designated G3, had a depth of 3.57 m, whereas G11 was the deepest at 41.57 m. As steepland gullies develop, their length, width, depth, area, and volume increase (Table 1).

An empirical model, $V = 0.2 \times L^{2.61}$, which accurately describes the relationship between the length and volume of the steepland gullies in the dry valleys, was established. This model exhibits a high coefficient of determination ($R^2 = 0.98$), indicating a strong correlation between the length and volume (Fig. 5A). The area and volume of the steepland gullies were found to increase nonlinearly with the length of the main gully (Figs. 5A and 5B). During steepland gully development, the longitudinal and transverse forces are exerted, leading to continuous growth, widening, and increased erosion amount. In the early stages of development, expansion is slow; however, in the later stages, the expansion rates accelerate rapidly. The gully elongation increases linearly with the width, length, and depth of the gully (Figs. 5C, 6A and 7A), where ductility values remain below 0.618 according to the Huang Jinlv's proportion standard, classifying it as a slender gully type. The gully openness also increases linearly with the width, length, and depth of the gully (Figs. 5D, 6B and 7B). Based on the openness degree, gullies G1–G7 were found to correspond to the incised type ($K < 0.35$), whereas gullies G8–G11 were found to correspond to the semi-open type ($0.35 < K < 0.65$). These results show that, as gullies develop, the amount of undercutting continues to increase as a result of hydraulic action. The vertical gradient decreases nonlinearly with the width, length and depth of the gully (Figs. 5E, 6D and 7D), and the coefficient of the main and tributary gullies decreases nonlinearly with the width, length, and depth (Figs. 5F, 6C and 7C). These two results indicate that the early stages of

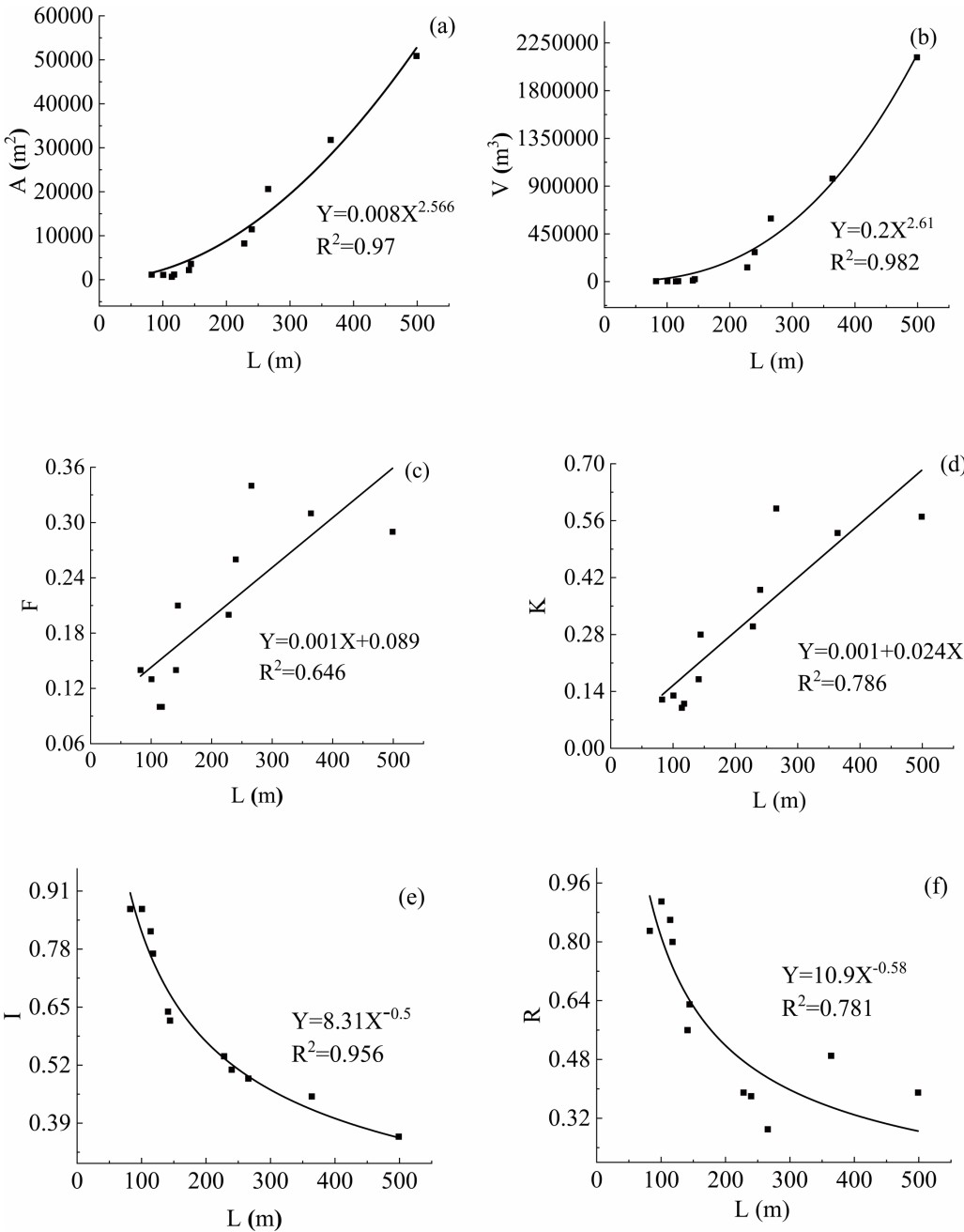

**Figure 5** **Correlation between gully length and other characteristic parameters.** (A) The correlation between gully length and gully area; (B) The correlation between gully length and gully volume; (C) The correlation between gully length and gully elongation; (D) The correlation between gully length and gully openness; (E) The correlation between gully length and vertical gradient; (F) The correlation between gully length and the coefficient of the main and tributary gullies.

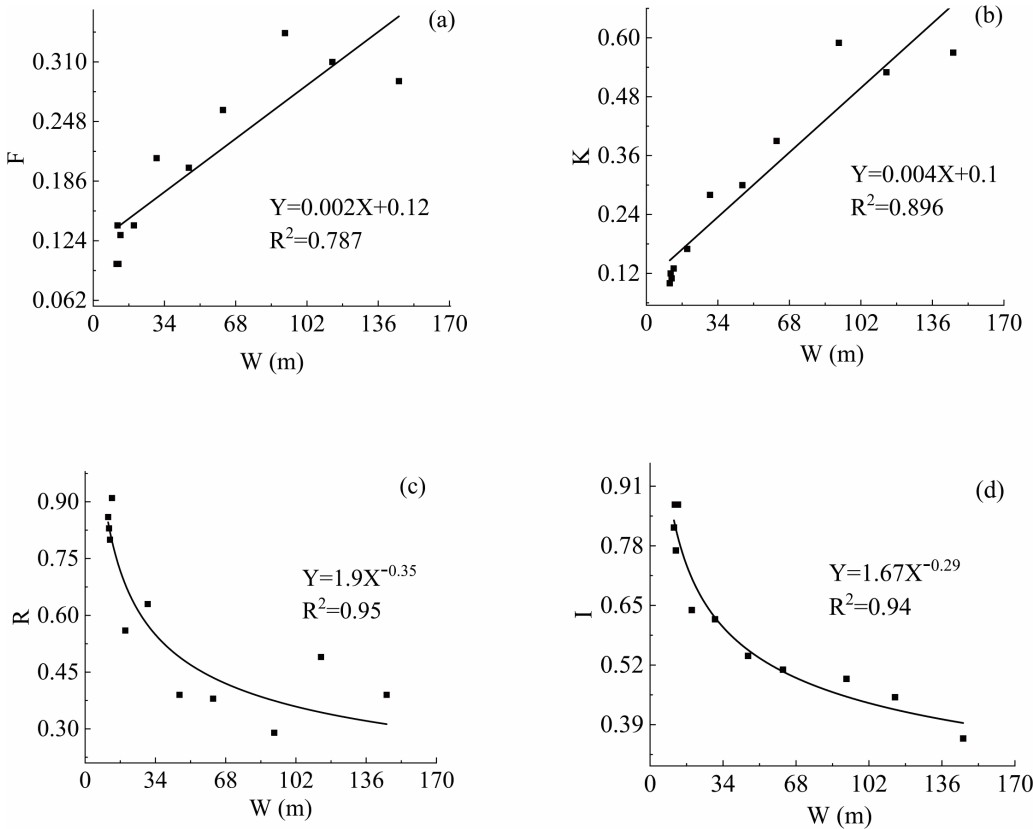

**Figure 6 Correlation between gully width and other characteristic parameters.** (A) The correlation between gully width and gull elongation; (B) The correlation between gully width and gully openness; (C) The correlation between gully width and the coefficient of the main and tributary gullies; (D) The correlation between gully width and vertical gradient.

gully development are rapid; the rate of development later slows down until it becomes constant.

### Profile characteristics

Based on the actual situation in the study area, four representative gullies were selected, and the five cross-sections described in "Steepland gully cross-section indicators" were drawn for each of them (Fig. 8); the average VF values for these cross-sections were then calculated. It was observed that the VF values gradually increased as the gullies developed (Fig. 9). The range falls between 0.38 and 1.35, where G1–G9 represent V-shaped gullies (VF < 1), while G10 and G11 represent U-shaped gullies (VF > 1).

As illustrated in Fig. 8, the cross section of the gully exhibits a gradual transition from wide to narrow, which is indicative of a wide gully head and a narrow gully tail. The four selected gullies have undergone progressive deepening and widening over time, transitioning from shallow V-shaped gullies to having a more pronounced U-shaped morphology. Initially, the cross-sections exhibit distinct V-shapes with smooth walls, primarily due to hydraulic scouring and undercutting. Subsequently, the gullies gradually

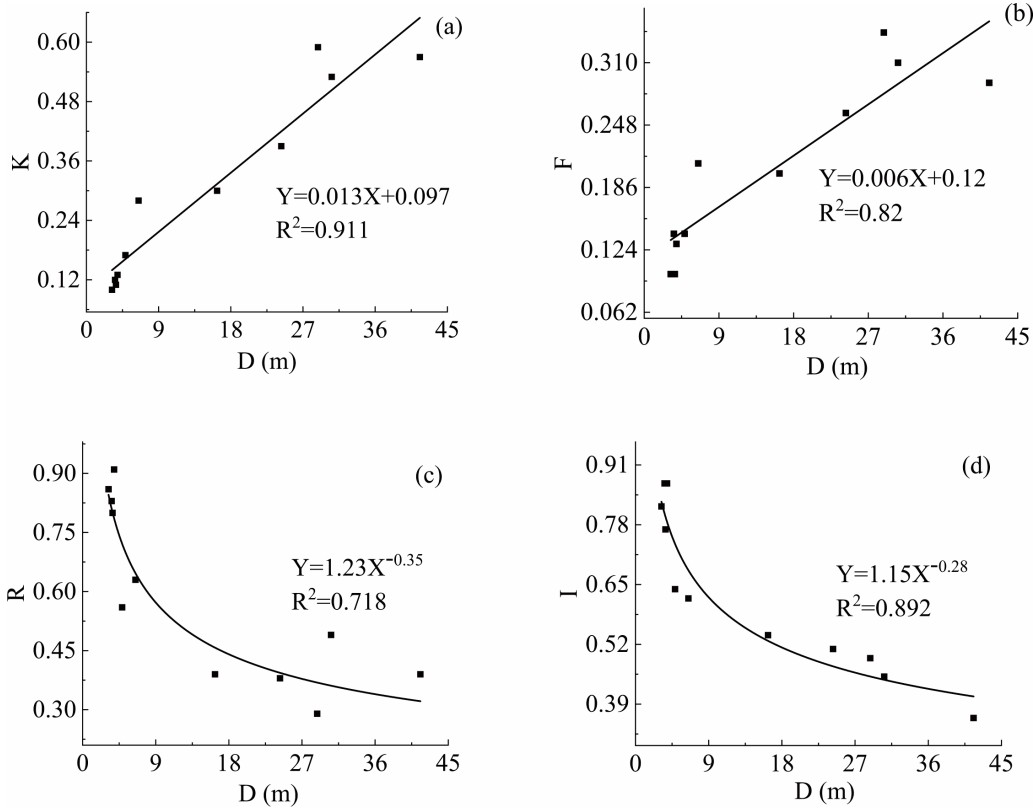

**Figure 7 Correlation between gully depth and other characteristic parameters.** (A) The correlation between gully depth and gully openness; (B) The correlation between gully depth and gully elongation; (C) The correlation between gully depth and the coefficient of the main and tributary gullies; (D) The correlation between gully depth and vertical gradient.

evolve into broader V-shaped gullies characterized by the formation of noticeable fractures and steps on their slopes; gravity-induced collapse also becomes increasingly prominent during this stage. At the same time, lateral erosion continues to widen the gullies, and surface tributaries develop. This leads to the slopes becoming less steep until stability is achieved within the U-shaped gullies.

## Division of the development of steepland gullies into stages

The steepland gully classification results show that each development stage is associated with different morphological characteristics (Tables 1 and 2). Gullies G1, G2, and G3 have lengths of between 82.29 and 114.01 m, widths of between 11.12 and 12.96 m, and depths of between 3.19 and 3.88 m. These gullies have VF values of less than 1 and V-shaped cross-sections. The scale of these gullies is small: they consist of a single gully with regular and single shape and are at the infant stage.

Gullies G4, G5, and G6 have lengths between 117.69 and 144.06 m, widths between 12.01 and 30.24 m, and depths between 3.68 and 6.50 m. These gullies have VF values of less than 1 and V-shaped cross-sections. The gully edges are severely broken. These gullies

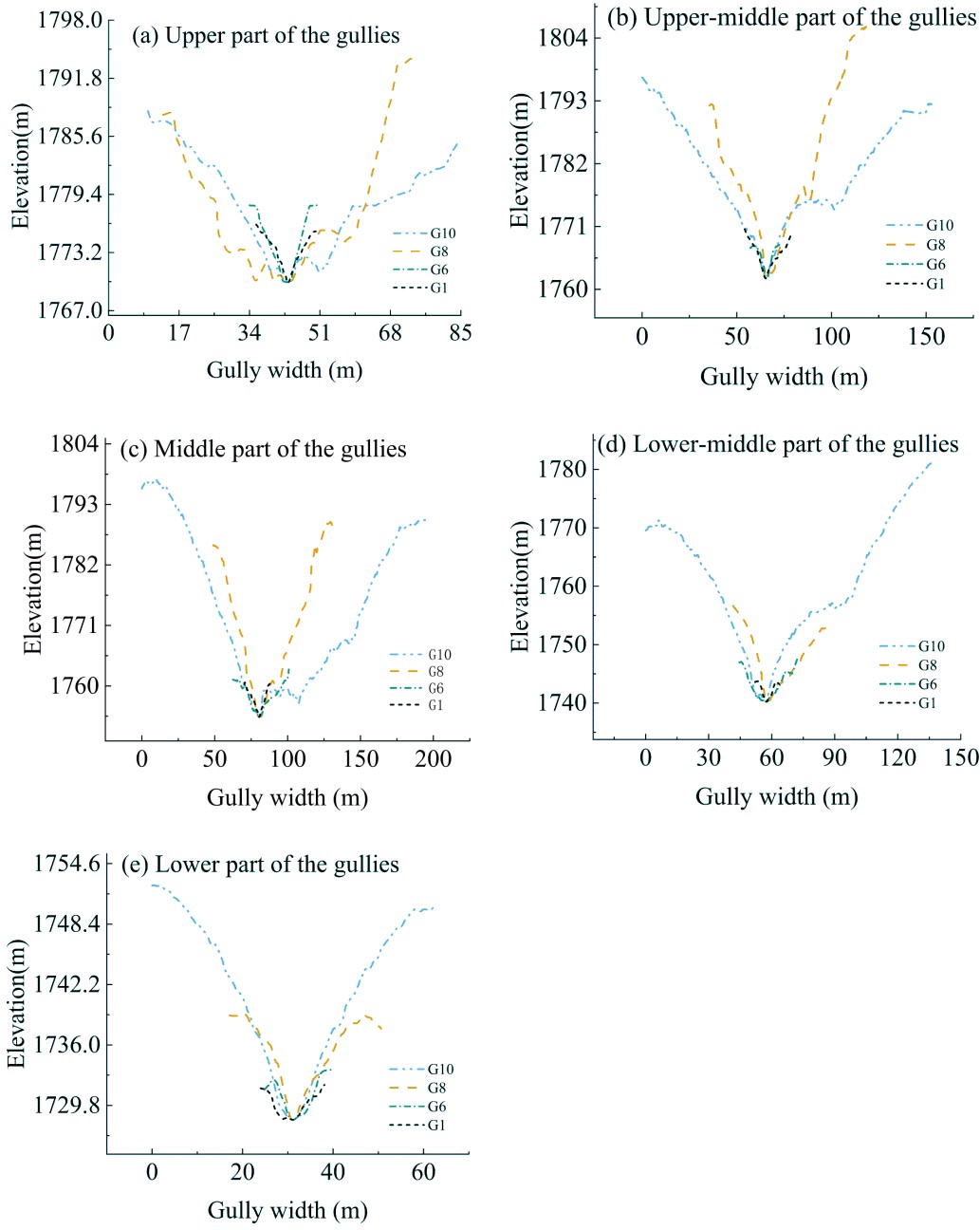

**Figure 8** Cross-section variations in various parts of gullies G1, G6, G8 and G10.

are spoon-shaped with a big top and small bottom, are irregular in shape, and are gradually expanding up to the gully wall; this results in an increased eroded area.

Gullies G7, G8, and G9 have lengths between 227.94 and 265.60 m, widths between 45.61 and 91.50 m, and depths between 16.29 and 28.86 m; they have VF values of less than 1. These gullies have wide V-shaped cross-sections and exhibit a further increase in gully

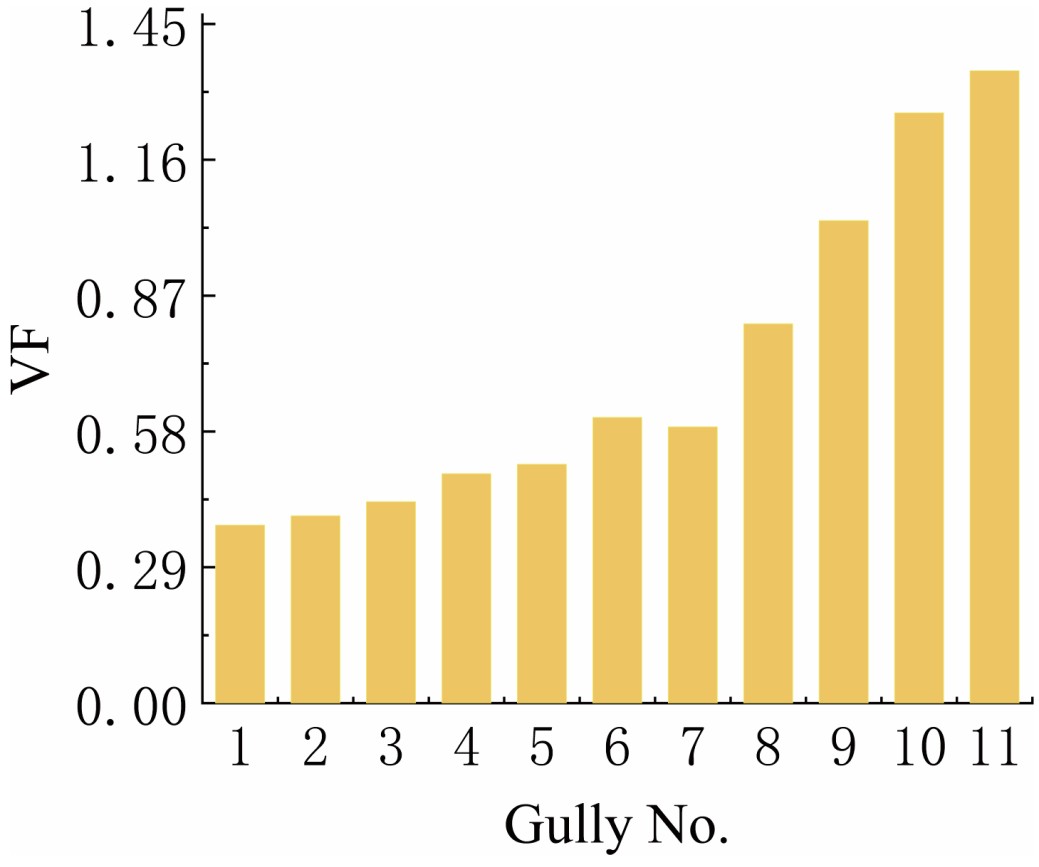

**Figure 9** **The values of width-to-height ratio (VF) of steepland gullies.**

scale as the gully heads collapse and move upstream; these gullies are at the mature stage and exhibit strong erosion.

Gullies G10 and G11 have lengths of 265.60 and 498.82 m, widths of 91.50 and 145.87 m, and depths of 30.57 and 41.57 m, respectively; for both gullies, VF > 1. These gullies have U-shaped cross-sections, and their internal branches are well developed. The head of the gully headward erodes the sloping farmland. Although the shape of this type of erosion gully can be irregular, gullies of this type are generally spoon-shaped. Later, during the old-age stage of steepland gully development, hydraulic erosion predominates due to the overall decrease in erosion potential energy.

## DISCUSSION

### The V–L empirical relationship of steepland gullies

The relationship between gully volume and gully length can be described by a power function, and numerous scholars have established a significant empirical V–L power function relationship to predict both the volume of gullies and the amount of erosion (*Caraballo-Arias et al., 2016*; *Dong et al., 2014*; *Frankl et al., 2013b*; *Wu et al., 2018*). The coefficients of the regression equations in different regions vary significantly due to

**Table 2  Morphological characteristics of steepland gullies at different development stages.**

| Steepland gully numbers | Development stage | Characterization | Profile and planar morphology |
|---|---|---|---|
| G1, G2, G3 | Infancy | The slope bedrock in the study area consists mainly of siltstone with developed bedding and joints, resulting in broken rocks. Surface runoff scours away a significant amount of clastics from the loose surface, causing steps to form on the slope. Numerous erosion breakpoints form along the longitudinal section of the runoff route, which develop into gully heads. As a result of the action of flowing water, continuous headward erosion occurs at these gully heads, leading to the lengthening of the rills. Eventually, many gully heads become connected, forming the initial steepland gully. |  |
| G4, G5, G6 | Youth | The erosion process has intensified and is dominated by hydraulic erosion, resulting in the steepland gully eroding downstream with further increases in length, width, and depth. |  |
| G7, G8, G9 | Maturity | The effect of gravity reduces the capacity of the surface runoff to transport deposits. In addition, transported deposits gradually accumulate from the head of the gully to its tail, leading to the formation of a steepland gully with a large head, wide body, and narrow tail. |  |
| G10, G11 | Old | The gully bed undergoes longitudinal erosion as a result of rainfall, resulting in the formation of a hydraulic drop. At the same time, headward erosion continuously erodes the gully head, causing its collapse, together with upstream movement along the slope direction. This process leads to an increase in the length, width, and depth of the gully while encouraging surface runoff. Finally, surface runoff occurs across the gully, which leads to a further decrease in the gully gradient. As a result, the gully gradually stabilizes, and the vegetation within the gully begins to recover. |  |

various influencing factors, including development stage, gully count, morphological characteristics, and measurement methods (*Li et al., 2017*; *Wu et al., 2018*). Therefore, establishing an empirical relationship between the length and volume of steepland gullies in the dry valleys can serve as a reference for the rapid estimation of gully volume. The statistical data of 11 steepland gullies were utilized in this study to establish power functions for length and volume. The regression equation $V = 0.2 \times L^{2.61}$ was derived, yielding a high coefficient of determination ($R^2 = 0.98$). These findings support previous research results

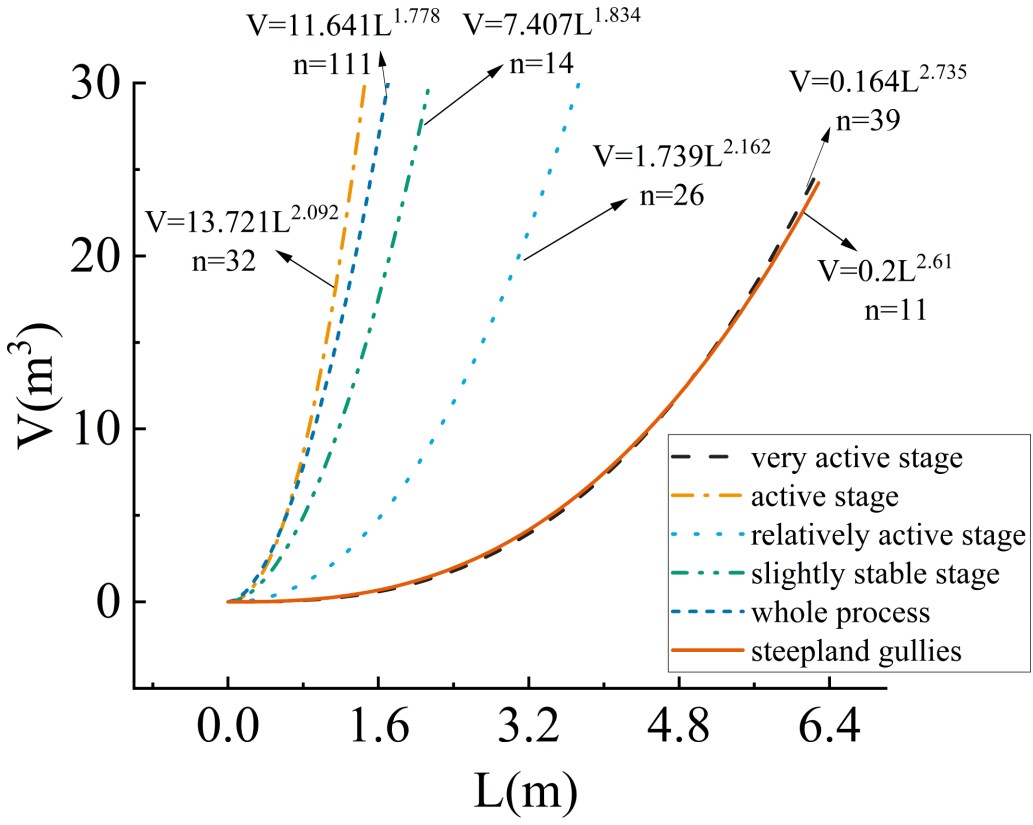

**Figure 10  Comparison of V–L relationship between dry valley gullies and steepland gullies.**

(*Dong et al., 2014*; *Zhang et al., 2022*), indicating a robust association between steepland gully length and volume. The conclusion is drawn that the steepland gullies belong to the gully category and are in the initial developmental stages. Compared to the empirical model proposed by *Yang et al. (2021a)*, the very active stage is characterized by $V = 0.164 \times L^{2.73}$ ($n = 39$), the active stage is characterized by $V = 13.721 \times L^{2.092}$ ($n = 32$), and the relatively active stage is characterized by $V = 1.739 \times L^{2.162}$ ($n = 26$). The slightly stable stage is represented by $V = 7.407 \times L^{1.834}$ ($n = 14$), while in the entire process, a moderately stable stage exists, with $V = 11.641 \times L^{1.778}$ ($n = 111$). Comparative analysis revealed that the steepland gullies are currently in a very active stage of dry valley gully development (Fig. 10). As the steepland gullies progress, both their length and width increase, while internal branches gradually form within the gully. Traceability erosion diminishes the gully watershed, causing it to merge with other gullies and expand continuously, ultimately transforming into a continuous gully system. Therefore, steepland gullies likely represent an early stage in the development of gully systems in the dry valley region of southwest China.

### Characteristics of steepland gully development in the dry valley

A steepland gully represents the initial stage of the formation of a permanent gully. The development of steepland gullies exhibits particular characteristics influenced by natural

factors, such as geology, topography, climate, soil, vegetation, and human activity. This results in distinctive developmental characteristics. The morphology of a steepland gully resembles that of a spoon gully found on the Loess Plateau of northern China. Despite both types exhibiting prominent gully lines with a wide head and narrow tail, taking on shapes resembling spoons or palms, the key difference is that the spoon gully's tail gradually tapers and either merges into a larger gully or directly disappears on the slope surface (*Heede 1982*; *Li et al., 2020*; *Li et al., 2017*). The Loess Plateau experiences limited rainfall and is dominated by cotton loess and sandy loess slopes. These slopes have high collapsibility and are highly susceptible to subsurface erosion of the loess; as a result, kettle depressions and hidden caverns, as well as spoon gullies, are formed (*Li et al., 2003*; *Li et al., 2020*; *Su et al., 2015*). In the dry valleys of southwest China, active geological structures, concentrated precipitation, sparse vegetation, and fragmented rock and soil provide the necessary conditions for steepland gully development. The slope experiences the formation of a series of water flow pits due to the combined effects of intense rainfall and gravity, leading to water-induced continuous headward erosion and ultimately initiating the development of steepland gullies (*Xiong et al., 2014*; *Zhao et al., 2013*). The joint action of gravity erosion and hydraulic erosion affects two types of gullies, with the headward erosion at the gully head markedly contributing to the promotion of further gully development.

Steepland gullies are widely distributed worldwide and are found in locations such as New Zealand and Ethiopia. These gullies form on steep slopes and are significantly influenced by precipitation. However, their development patterns and morphology vary due to differences in climate and lithology. In the dry valleys of southwest China, the formation of steepland gullies can be attributed to heavy rainfall. Once they have formed, these gullies rapidly erode the bedrock, leading to collapses that damage vegetation and arable land. In the study area in New Zealand, steepland gullies form as a result of deforestation and intensive agriculture. Typically triggered by landslides, these gullies develop in areas where multiple drainage channels converge; these are subsequently carved out and deepened by flowing water. The rate at which these gullies develop and their development patterns are primarily influenced by the steep topography and underlying lithology (*Marden et al., 2012*). In the study area in Ethiopia, the natural vegetation has been destroyed due to human activities such as grazing and crop planting. On the hillsides, this has resulted in steepland gullies caused by rainfall (*Frankl et al., 2013a*). Regarding their morphological characteristics, the steepland gullies in the dry valleys of southwest China and those in New Zealand exhibit similar features, such as a wide head and a thin tail that resembles a spoon or a palm. However, research on the erosion of steep slopes in Ethiopia revealed that a high clay content and the presence of extensive rock fragments in the soil on these slopes increases the susceptibility to sheet erosion when the slopes are exposed to rainfall (*Nyssen et al., 2006*).

## CONCLUSIONS

In this study, high-resolution DEM images acquired by unmanned aerial vehicles were used to select 11 steepland gullies in the dry valley region of southwest China based

on space-for-time substitution. The length, width, depth, area, volume, coefficient of main and tributary gullies, vertical gradient, gully elongation, gully openness, and the width-to-height ratio of the gullies were extracted using ArcGIS software. By considering the morphological characteristics of the different stages of gully development, this development was divided into four stages. Quantitative analysis of both the planar and profile morphology of the gullies at these different stages was used to deduce how the evolution of the gullies progressed. The following two main conclusions can be drawn. (1) Based on the morphological characteristics of gully development, the evolution of steepland gullies can be categorized into four stages: infancy, youth, maturity, and old age. Each stage exhibits a distinct profile and planar features. The gullies designated G1, G2, and G3 were found to be in their infancy, gullies G4, G5, and G6 to be at the youth stage, and gullies G7, G8, and G9 to be mature gullies. Two gullies—G10 and G11—were found to have reached the old-age stage. Steepland gullies have their origin in intense rainfall events that result in scouring caused by slope runoff. Subsequent lateral and headward erosions together lead to the widening of the gully as the area of the adjacent watershed decreases; this leads to rapid development of the gully. In the later stages of development, gravity erosion dominates, although lateral erosion also occurs; however, due to the reduced overall erosion potential, the secondary erosion and transport resulting from hydraulic action ultimately stabilize the gully. (2) During the development of steepland gullies, there is a significant linear positive correlation between the gully's degree of openness and elongation with increasing gully length, width, and depth. The vertical gradient and the coefficient of main and tributary gullies also exhibit power-law relationships with these gully dimensions. The simultaneous occurrence of longitudinal and lateral erosion leads to continuous growth and widening of the gullies, resulting in an increase in the amount of erosion. Initially, the expansion of a steepland gully proceeds at a relatively slow pace with gradual increases in both area and volume; however, during the later stages, the development accelerates significantly, leading to a rapid expansion in both the area and volume of the gully. Nevertheless, as the gully continues to develop, its overall potential energy gradually diminishes causing the gully to tend toward stabilization. The V–L erosion empirical model for steepland gullies was established and compared with that of typical gullies in the dry valley region of southwest China. The findings indicate that the development law of steepland gullies is essentially consistent with the very active stage of typical gully formation, indicating that steepland gullies may represent an early stage in gully development. This model can effectively predict the volume and extent of erosion caused by steepland gullies in the dry valleys of southwest China, thereby providing valuable insights for the rapid estimation of cutting volumes.

### Funding

This work was supported by the Natural Science Foundation of Sichuan Province (Grant No. 2022NSFSC1734); the National Natural Science Foundation (Grant No. 41971015, 42307454); the Doctoral Research Initiation Program of China West Normal University

(18Q018, 20E030, 22kA002, 19E067) and the Innovation Team Funds of China West Normal University (KCXTD 2022-1). The funders had no role in study design, data collection and analysis, decision to publish, or preparation of the manuscript.

## Grant Disclosures

The following grant information was disclosed by the authors:

The Natural Science Foundation of Sichuan Province: 2022NSFSC1734.

The National Natural Science Foundation: 41971015, 42307454.

The Doctoral Research Initiation Program of China West Normal University: 18Q018, 20E030, 22kA002, 19E067.

The Innovation Team Funds of China West Normal University: KCXTD 2022-1.

## Competing Interests

The authors declare there are no competing interests.

## Author Contributions

- Tingting Cui performed the experiments, prepared figures and/or tables, and approved the final draft.
- Yuli He conceived and designed the experiments, authored or reviewed drafts of the article, and approved the final draft.
- Lei Wang conceived and designed the experiments, authored or reviewed drafts of the article, and approved the final draft.
- Jun Luo performed the experiments, authored or reviewed drafts of the article, and approved the final draft.
- Ting Xiao performed the experiments, prepared figures and/or tables, and approved the final draft.
- Hui Liu performed the experiments, analyzed the data, authored or reviewed drafts of the article, and approved the final draft.
- Bin Zhang conceived and designed the experiments, authored or reviewed drafts of the article, and approved the final draft.
- Qingchun Deng analyzed the data, authored or reviewed drafts of the article, and approved the final draft.
- Haiqing Yang analyzed the data, authored or reviewed drafts of the article, and approved the final draft.

## Data Availability

The raw data are available in the Supplementary File.

## Supplemental Information

Supplemental information for this article can be found online at http://dx.doi.org/10.7717/peerj.18411#supplemental-information.

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
