# Peer review of "Characteristics and development of steepland gullies in the dry valleys of Southwest China"

_PeerJ, doi:10.7717/peerj.18411_

## Round 0.1 · original submission · Major Revisions

We have received two reviews with differing assessments of your submission. Please address the comments from both reviewers, with particular attention to the points raised by Reviewer 1, and resubmit your manuscript for further consideration.

Reviewer 1 ·

Basic reporting

Dear Authors,

I have reviewed your manuscript on the development of spoon gullies in Southwest China. While the paper's title suggests it would be of broad research interest, the content of the manuscript falls short of several standards and does not align with the title as it stands. The main issues preventing publication are as follows:

1) Lack of clarity in the study's objectives and design
2) Limited number of gully catchments, hindering the ability to draw general conclusions based on gully morphology
3) Inappropriate structure of the results and insufficient discussion
4) Lack of novelty

Experimental design

The first major issue is the unclear description and explanation of what constitutes a spoon gully. According to Li et al. (2002), “…a spoon gully is in the upper or middle part of a hillslope and not connected to the existing gully system.” The gullies you selected seem quite different in terms of connectivity: gullies G1, G3, and G4 are disconnected systems and are typical examples of spoon gullies, while gullies G5–G11 are connected systems at different stages of development. Can these gullies be characterized as spoon gullies?

Even if we accept that all the gullies can be analysed, the limited number of gullies (eleven) is not representative enough to draw general conclusions about the characteristics of gully (or spoon gully) systems based on morphology. Studies with similar approaches typically cover a broader region and analyse a greater number of gully systems (compare your research design with the papers by Li et al. (2020, 2022) and Zhang et al. (2022)). Such a localized analysis does not provide new evidence about spoon gullies, resulting in a poor discussion, which I will address further.

Validity of the findings

I did not find any novelty or identification of a research gap in this study. It appears to be an overall introduction to (spoon) gully development with a local case study showing different stages of gullies, but it lacks additional value in terms of age (dating methods) or potential development (using modelling approaches). The “static” geomorphometric analyses require much more input data to be of regional or global importance to the research community.

The results section is a mixture of introduction (the beginnings of sections 4.1.1 and 4.1.2 contain references) and interpretations (more suitable for the discussion section). Chapter 4.3 does not present any results—it is rather a general summary of what is known about the development of (spoon) gullies. Therefore, it is not surprising that the discussion does not bring any new aspects to your research. The first part is more about the study site, while the latter part follows section 4.3 without any implications from your findings.

Additional comments

I do not wish to be overly critical of your work, as the findings you presented may serve as an important basis for future research. However, in its current state, these findings are insufficient for publication in a high-impact journal. I recommend either extending your focus area to include many more gullies to find any general context of gully development stages or focusing more specifically on gully specifics (e.g., numerical dating of gully development or modeling the evolution of selected gully formations).

·

Basic reporting

In line 62, a space should be inserted between the word observed and the literature citation.
Line 331, delete the superscript number above the word Loess Plateau.

Experimental design

No comment.

Validity of the findings

No comment.

Additional comments

Your introduction needs more detail. I suggest that you improve the description by adding references that talk about the same topic from other parts of the world (for example Ionita et al., 2021; Le Roux, Sumner 2011; Martins et al., 2022; Rahmati et al., 2022).
The English language is satisfactory and understandable for an international audience.
The methodology has already been used in other manuscripts and is clearly explained.
The tables and figures are clear and appropriate. The correlations are quite high. Based on the results and correlations obtained, the authors have good conclusions in the manuscript. The objectives of the study were answered.
Strengthening of the manuscript: Field research and remote sensing measurements were used in the manuscript, which certainly contributes to a better understanding of the problem of the occurrence and development of spoon gullies.
The only weakness of the manuscript is that you have not compared your conclusions and discussions with other parts of the world. Is there anywhere else in the world, at least to a lesser extent, the possibility of the occurrence of this kind of gullies (other than the Loess Plateau), how did they form and develop there?
The literature is new, up-to-date, and well-cited.

---

## Round 0.2 · Minor Revisions

I agree with the reviewers’ comments that the manuscript has improved after revision. However, further restructuring and reorganization would enhance the clarity and flow of the paper. Specifically, the Results section should focus solely on presenting the authors’ findings, which are typically concise statements derived from the work itself. Any discussion, speculation, or preferred explanations should be moved to the Discussion section, where they can be properly elaborated.

Reviewer 1 ·

Basic reporting

Dear authors,

Thank you for the significant improvements made to your revised manuscript. While I still have some concerns regarding the novelty and added value of the study compared to previously published papers on gully development in similar environments, I acknowledge the considerable effort put into enhancing the original version. That being said, the primary issues remain, particularly with the structure of the results and discussion sections, as well as the need for a clearer, more logical presentation of the calculated V-L relationship. Please see my comments below:

Section 4.2: This section needs to be more concise. Please focus on your key results, and move any broader implications to the discussion or consider removing them. Much of the information is general and does not add significant value. For instance, lines 267–279, which describe the development of gullies, should be condensed and, if relevant to your findings, discussed in the discussion section. As it appears that these points are derived from other studies rather than your own results (which were primarily based on geomorphometric analyses), I suggest revising the structure accordingly and applying the same approach to other stages of gully development described later.

Section 5.1.1: The new information needs to be organized more clearly. I recommend creating a new section in the methods that describes the V-L relationship and explains its relevance to your research. Then, present the results of this relationship in the results section. Finally, discuss your findings in comparison to other studies (as has been partially done in Section 5.1.1). In the abstract, please spell out the full name of the V-L relationship (rather than using the abbreviation) and introduce it more effectively in the methods.

Section 5.1.2 (from line 362 onwards): Start a new paragraph here. The first part, discussing steepland gullies, would be more appropriate in the introduction (focusing on the occurrence of steep gullies in different parts of the world). The second part, from around line 371 onwards, fits better in the discussion. Please extend this section by comparing your findings with studies from other regions, such as loess gullies in Poland or steepland gullies in various parts of the world. I recommend referencing studies such as Frankl et al. (2013) – as you have already us it – or Yibetal’s work from Ethiopia. But there are a lot of others.

Figures 5, 6, 7: Please provide more detailed captions for each figure, including explanations of each variable. The current figure captions only use letters to label the axes, but the variables should be named clearly in the captions. Ensure consistency across all figures.

Figure 9 – Please include the meaning of "VF" in the figure caption to make it more informative. Currently, the explanation is only provided in the methods section.

Experimental design

-

Validity of the findings

-

Additional comments

-

·

Basic reporting

No comment.

Experimental design

After correcting the work, the aims of the work are clearly defined. The methodology is appropriate now that the work title has been corrected.

Validity of the findings

No comment.

---

## Round 0.3 · accepted · Accept

Dear Cui,

I am pleased to inform you that your manuscript titled “Characteristics and development of steepland gullies in the Dry valleys of Southwest China” has been accepted for publication in PeerJ. After thorough review and consideration, our editorial team and reviewers have found your work to be a valuable contribution to the field.

We appreciate the effort and dedication you have put into your research and manuscript preparation. Your findings provide significant insights and advancements that will benefit the scientific community.

Once again, congratulations on the acceptance of your manuscript. We look forward to seeing your work published in PeerJ and to your continued contributions to the field.

Best regards,

Chris Zou
Associate Editor of PeerJ